# *Enterococcus faecalis* Shields *Porphyromonas gingivalis* in Dual-Species Biofilm in Oxic Condition

**DOI:** 10.3390/microorganisms10091729

**Published:** 2022-08-27

**Authors:** Huan Chang Tan, Gary Shun Pan Cheung, Jeffrey Wen Wei Chang, Chengfei Zhang, Angeline Hui Cheng Lee

**Affiliations:** Division of Restorative Dental Sciences, Faculty of Dentistry, University of Hong Kong, Pokfulam, Hong Kong SAR, China

**Keywords:** aerobic conditions, biofilm model, confocal laser scanning microscopy, cell viability

## Abstract

Aim: To develop a reproducible biofilm model consisting of *Enterococcus faecalis* (*E. faecalis*) and *Porphyromonas gingivalis* (*P. gingivalis*) and to evaluate the interaction between the two bacterial species. Methodology: *E. faecalis* and *P. gingivalis* were grown in mono-culture, sequential, and co-culture models for 96 h in a 96-well polystyrene microtiter plate under both aerobic and anaerobic conditions separately. The viability of the two bacterial species in the biofilms was quantified by polymerase chain reaction (qPCR). Biofilm thickness and protein contents were measured using confocal laser scanning microscopy (CLSM). Two-way analysis of variance (ANOVA) was performed to analyze cell viability and biofilm thickness among different culture models cultivated under either aerobic or anaerobic conditions. The level of significance was set at *p* < 0.05. Results: Different culture models tested did not show any significant difference between the viable cell counts of both *E. faecalis* and *P. gingivalis* cultivated under aerobic and anaerobic conditions (*p* > 0.05). Biofilm was significantly thicker (*p* < 0.05) in the co-culture models compared to the mono-culture and sequential models. Protein contents in the biofilms were more pronounced when both bacterial species were co-cultured under aerobic conditions. Conclusions: *E. faecalis* appeared to shield *P. gingivalis* and support its continued growth in oxic (aerobic) conditions. The co-culture model of *E. faecalis* and *P. gingivalis* produced a significantly thicker biofilm irrespective of the presence or absence of oxygen, while increased protein contents were only observed in the presence of oxygen.

## 1. Introduction

The oral cavity is a well-oxygenated environment, where the elevated oxygen tension modulates oxygen-tolerating bacterial cells to increase their enzymatic and non-enzymatic reduction of molecular oxygen to superoxide anions. This results in the formation of hydrogen peroxide and oxygen by dismutation, and the former reacts with superoxide anions to form hydroxyl ions in the presence of iron complexes [1]. These free oxygen radicals are highly reactive and able to cleave nucleic acids and oxidize essential proteins and lipids [2,3]. In contrast, obligate anaerobic microorganisms, such as *P. gingivalis*, do not possess the mechanism of anti-oxidation. Thus, the reactive oxygen species must be detoxified to minimize the undesirable effects on the obligate anaerobic microorganisms.

*P. gingivalis,* a group of obligate anaerobic bacteria, is highly associated with periodontal diseases, ranging from reversible gingivitis to irreversible periodontitis [4]. Along with *Tanneralla forsythia* (*T. forsythia*) and *Treponema denticola* (*T. denticola*), they form the “red complex” and have been implicated as the primary causative pathogens in periodontal diseases [5,6]. They progressively degrade the periodontal collagen and non-collagenous tissues, forming periodontal pockets [7]. *Enterococcus faecalis* (*E. faecalis*) is the most frequently isolated bacterial species from symptomatic root canal-treated teeth, with reported prevalence in up to 90% of cases [8,9,10]. *E. faecalis* is often found in secondary or persistent cases due to its ability to survive in harsh environments with nutrient deprivation and high alkalinity despite the presence of intracanal medicaments [11,12,13,14]. The pathogenicity and difficulty of their eradication have been attributed to the ability of *E. faecalis* to form biofilms, which can be 1000-fold more resistant to antimicrobials than their planktonic counterparts [15,16]. Both *E. faecalis* and *P. gingivalis* have been frequently isolated in infected root canals and periodontal pockets in failed cases by culturing and molecular identification techniques [17,18].

*P. gingivalis* can tolerate low levels of oxygen (6–10%) and reach a steady state of growth depending on the availability of hemin [19,20]. *Fusobacterium nucleatum* (*F. nucleatum*), another obligate anaerobic bacteria, supports the growth of *P. gingivalis* in an unfavorable oxygenated and carbon dioxide-depleted environment [19,21]. *F. nucleatum* creates a microenvironment with reduced oxygen tension that protects *P. gingivalis* within the niche [19]. A recent study demonstrated the potential of *Candida albicans* (*C. albicans*), a yeast-like fungus, in protecting anaerobic bacteria in an aerobic environment [22]. As a facultative anaerobe, *C. albicans* creates a hypoxic microenvironment within the fungal biofilms, which is conducive to the growth of *P. gingivalis* [22]. To date, there is no research conducted to study the interaction between *E. faecalis* and *P. gingivalis* when they co-exist. Therefore, this study aimed (i) to develop a simple and reproducible biofilm model consisting of *E. faecalis* and *P. gingivalis* and (ii) to study the interaction between these two bacterial species under both aerobic and anaerobic conditions separately, either using single- or dual-species culture models.

## 2. Materials and Methods

### 2.1. Bacterial Strains and Culture Conditions

*E. faecalis* (ATCC 47077) and *P. gingivalis* (ATCC 33277) were used in this study. The strains were stored at −70 °C as freeze-dried cultures and recovered on blood agar supplemented with 40 mL of horse blood and 8 mL of Hemin-Vitamin K solution (Sigma-Aldrich, Burlington, MA, USA) before cultivation in Brain-heart Infusion Broth (BHI) (Thermo Fisher Scientific™, Oxoid, UK). *E. faecalis* was cultivated under aerobic conditions at 37 °C for 24 h, while *P. gingivalis* was cultivated under anaerobic conditions (85% N_2_, 10%H_2,_ and 5% CO_2_) at 37 °C for 24 h. For dual-species biofilm (sequential and co-culture) development, both bacterial species were cultivated under aerobic or anaerobic conditions separately and independently for 48 h, with media replenished every 48 h before subjecting the biofilm to further culturing under a different condition.

### 2.2. Development of Biofilm and Incubation

Cultures of *E. faecalis* and *P. gingivalis* were centrifuged and washed with phosphate-buffered solution (PBS) after the supernatant removal. Fresh Pg Broth (TSB, Sigma Aldrich; Yeast extract, Thermo Fisher Scientific™) was added to the pellet. Bacterial density was adjusted to 0.271–0.279 (2 × 10^8^ CFU/mL) using a spectrophotometer (Beckman Coulter DU530 Life Science UV/Vis Spectrophotometer, Brea, CA, USA) at an optical density of 1 and a wavelength of 660 nm, according to MacFarland Standard scale no. 2. Then, 10-fold serial dilutions of the bacterial suspension were made up to 10^−2^ in tubes containing sterilized Pg broth. A total of 200 µL of the bacterial suspension was deposited into 96-well microtiter plates in quadruplicates, and 200 µL of sterilized Pg broth was deposited into the same well plates in triplicates to serve as the negative control. The same procedure was repeated in another two microtiter plates. For dual-species cultures, i.e., sequential or co-culture models, an equal volume of the two mono-species cultures were combined. All the specimens were incubated either aerobically or anaerobically for 96 h, and the culture medium was replenished after the first 48 h.

### 2.3. Experimental Groupings

Three different models of dual-species biofilm were developed in the flat-bottomed 96-well polystyrene microtiter plates in quadruplicates. The experimental culture models were divided into the following groups:

Sequential model I: *E. faecalis* biofilm was formed by seeding bacterial cells into the microtiter plate wells (100 µL; 1 × 10^6^ cells/well) in Pg broth for 48 h of aerobic incubation. Then, the medium was removed and replenished. *P. gingivalis* (100 µL; 1 × 10^6^ cells/well) were added and cultured for another 48 h under aerobic and anaerobic conditions separately. After the supernatant removal, each well was washed with 200 µL of PBS for further analysis.

Sequential model II: *P. gingivalis* biofilm was formed by seeding bacterial cells into the microtiter plate wells (100 µL; 1 × 10^6^ cells/well) in Pg broth for 48 h of anaerobic incubation. The medium was then removed and replenished. *E. faecalis* (100 µL; 1 × 10^6^ cells/well) were added and cultured for another 48 h under aerobic and anaerobic conditions separately. Afterward, the supernatants were removed. Each well was washed with 200 µL of PBS for further analysis.

Co-culture model: Both *E. faecalis* and *P. gingivalis* were grown simultaneously for 96 h under aerobic and anaerobic conditions separately. *E. faecalis* (100 µL; 1 × 10^6^ cells/well) and *P.gingivalis* (100 µL; 1 × 10^6^ cells/well) cells were seeded into the microtiter plate at the same time and then incubated for 96 h under aerobic and anaerobic conditions separately. The medium was refreshed after the first 48 h. After 96 h, each well was washed with 200 µL of PBS for further analysis.

Positive control: In parallel with each of these dual-species biofilm models, *E. faecalis* (200 µL; 1 × 10^6^ cells/well) and *P. gingivalis* (200 µL; 1 × 10^6^ cells/well) were cultured as mono-species under aerobic and anaerobic conditions separately to serve as positive controls.

All experiments were repeated on three independent occasions. 

### 2.4. Quantitative Analysis of Bacterial Cell Viability

After removing the supernatant, biofilms were washed with 200 µL of PBS, detached using the pipette tips, and transferred to the 1.5 mL Eppendorf tubes (SARSTEDT AG & Co. KG, Sarstedtstraß, Nümbrecht, Germany). The biofilm was re-suspended in 1 mL PBS and divided into two groups. One group was used for Propidium monoazide (PMA) staining (PMAxx™, Biotium, Fremont, CA, USA), and another group was used for DNA extraction.

For PMA staining, 1.25 µL of PMA was added to the biofilm suspension and left on ice for 10 min. Subsequently, samples were exposed to PMA-Lite LED Photolysis Device (Biotium Inc., Fremont, CA, USA) for 5 min and subjected to further centrifugation to remove the supernatant. The DNA was extracted using the QIAamp DNA Mini Kit (Qiagen, Hilden, Germany) according to the manufacturer’s instructions. Compositional analyses were enumerated using viability PCR (v-PCR). In brief, 1 µL of extracted DNA was added to 10 µL TaqMan Mix (Life Technologies, Thermo Fisher Scientific™, Waltham, MA, USA), MilliQ water (Merck Millipore, Burlington, MA, USA), and 1 µL of 10 µM forward/reverse primers (Life Technologies, Thermo Fisher Scientific™) and Taqman probes (Life Technologies, Thermo Fisher Scientific™) that were bacterial species-specific. The probes and primers are listed in Table 1. The thermal parameters used were as follows: (i) denaturation at 95 °C for 2 min; (ii) 40 cycles at 95 °C for 10 s; (iii) 58 °C for 30 s, using the StepOne plus Real-Time PCR System (Applied Biosystems™, Thermo Fisher Scientific™); and (iv) StepOnePlus software (Applied Biosystems™, Thermo Fisher Scientific™) for data compilation. Samples were quantified by calculating the colony-forming equivalent (CFE) based on an established standard curve of microbial colony-forming units ranging from 1 × 10^3^ to 10^8^ CFU/mL of *E. faecalis* and *P. gingivalis*, which DNA was extracted and processed with RT-PCR’s procedures. All samples were processed in duplicates in the v-PCR, with negative control samples containing water, primers, and master-mix only to rule out possible DNA contamination.

### 2.5. Biofilm Protein and Thickness Analysis with CLSM

The biofilm samples were stained with SyPRO biofilm matrix stain (Thermo Fisher Scientific™, Invitrogen™) and counterstained with Syto9 (Thermo Fisher Scientific™, Invitrogen™). In brief, planktonic cells were removed and washed gently with 200 µL of PBS. Biofilms were then stained, and the z-stacks were obtained from 5 different spots using CLSM (FV1000, Olympus, Shinjuku, Tokyo, Japan). The images were processed and viewed using FV10-ASW 4.2 (FV10, Olympus). Biofilm thicknesses were measured using the z-dimension of the CLSM images.

### 2.6. Data Analysis

All assays were carried out in three independent experiments, and the results were expressed as mean ± SD. Statistical analysis of the data was performed by two-way analysis of variance (ANOVA) using IBM SPSS Statistics version 26 (SPSS, Chicago, IL, USA). The level of significance was set at *p* < 0.05.

## 3. Results

### 3.1. Sequential Model I

The viability of *E. faecalis* in Sequential model I under aerobic or anaerobic conditions is similar to that of *E. faecalis* in monospecies culture (*p* > 0.05) (Figure 1a). The log10/mL values for *E. faecalis* under aerobic and anaerobic conditions were 8.30 and 8.22, respectively; in the positive control group, the values were 8.25 and 8.23, respectively (Figure 1a). The dual-species culture model and oxygen tension did not significantly alter the viability of *E. faecalis* (*p* > 0.05). The viability of *P. gingivalis* in Sequential model I under aerobic or anaerobic conditions is similar to that of *P. gingivalis* in monospecies culture (*p* > 0.05) (Figure 1b). The log10/mL values for *P. gingivalis* under the aerobic and anaerobic conditions were 3.99 and 5.59, respectively, whereas the values were 4.10 and 4.72 in the positive control group, respectively (Figure 1b).

### 3.2. Sequential Model II

Under aerobic or anaerobic conditions, the amount of viable *E. faecalis* in Sequential model II was similar to that of the *E. faecalis* as monospecies in the positive control group (*p* > 0.05) (Figure 2a). The log10/mL values for *E. faecalis* monospecies culture under aerobic and anaerobic conditions were 8.10 and 8.13, respectively, whereas the values for Sequential model II were 8.01 and 8.08, respectively (Figure 2a). The dual-species culture model and oxygen tension did not significantly alter the viable cell counts of *E. faecalis* (*p* > 0.05).

Under aerobic conditions, the amount of viable *P. gingivalis* in the Sequential model II was similar to that of the *P. gingivalis* as mono-culture in the positive control group (*p* > 0.05) (Figure 2a). Under anaerobic conditions, there was a 1-log reduction observed when *P. gingivalis* was grown in the Sequential model II without significant difference (*p* > 0.05) (Figure 2b). The log10/mL values for *P. gingivalis* as mono-culture under the aerobic and anaerobic conditions were 6.85 and 8.10, respectively, whereas the values for Sequential model II were 7.13 and 6.95, respectively.

### 3.3. Co-Culture Model

There was no significant alteration (*p* > 0.05) in the viable cell counts of *E. faecalis* and *P. gingivalis* in the co-culture models compared with their mono-culture counterparts. The log10/mL values for *E. faecalis* in mono-culture under aerobic and anaerobic conditions were 8.25 and 8.23, respectively, whereas the values in the co-culture model were 8.16 and 8.19, respectively (Figure 3). The log10/mL values for *P. gingivalis* in mono-culture under aerobic and anaerobic conditions were 4.10 and 4.72, respectively, whereas the values for the co-culture model were 4.44 and 3.74, respectively (Figure 3). Oxygen tension did not significantly impact the viable cell counts of these two bacterial species when they were cultured simultaneously (*p* > 0.05).

### 3.4. Biofilm Thickness

In Sequential model I, no significant difference was found in biofilm thickness (*p* > 0.05) between the mono-cultures and dual-species sequential models. The presence or absence of oxygen did not affect the biofilm formation. However, *E. faecalis* formed thicker biofilms than *P. gingivalis* in monospecies culture, i.e., 12.7 μm and 9.1 μm, respectively, under aerobic conditions (*p* < 0.05). For Sequential model I, a similar biofilm thickness in the dual-species sequential model was observed under aerobic and anaerobic conditions (Figure 4a,b).

As shown in Figure 5a,b, no significant difference in the biofilm thickness could be observed when the two bacterial species were grown sequentially in Sequential model II compared with their monospecies culture counterparts (*p* > 0.05). The presence or absence of oxygen did not affect the biofilm formation and thickness. In contrast, significant differences in the biofilm thickness were found in both the *E. faecalis* (*p* = 0.029) and *P. gingivalis* (*p* = 0.043) between their co-culture and mono-culture biofilm models under both aerobic and anaerobic conditions (Figure 6a,b). In fact, a significant increase in biofilm thickness was observed in co-cultures (*p* < 0.05) compared to the positive controls. When co-cultured aerobically, the biofilm formed was approximately one-fold thicker than their monospecies culture counterparts (Figure 6a,b).

### 3.5. Biofilm Protein Visualization

Under the aerobic condition, no difference in the protein contents was observed from the biofilms in the Sequential models I and II. The protein contents in the biofilms for both *E. faecalis* (Figure 7b) and *P. gingivalis* (Figure 7f) were more pronounced in the co-culture models compared with their mono-culture counterparts under aerobic conditions (Figure 7a,e). In anaerobic conditions, no visible difference was observed in the biofilm’s protein contents for both bacterial species in mono-culture and co-culture models (Figure 7c,d,g,h).

## 4. Discussion

A large body of evidence has suggested that bacterial behaviors in multi-species biofilm differ from the monospecies biofilm. Hence, research into the interactions between microbial species in biofilms can reveal the crucial factors that drive stability and changes leading to dysbiosis and the development of preventive and therapeutic approaches [23,24]. In vitro studies on dual- or multi-species biofilms have been carried out under controlled conditions to address the microbial community lifestyle in the root canal and periodontal pocket, assess their interactions and effects of stresses, as well as investigate and modify current therapeutic and preventive protocols [25,26,27]. This in vitro laboratory study employed dual-species bacteria, namely *E. faecalis* and *P. gingivalis*, to develop a simple and reproducible biofilm model. This dual-species biofilm model produced laboratory biofilm samples with minimal variation, which are readily assessable by microscopic and molecular analyses. A combination of techniques, including RT-PCR using viable cell staining and confocal laser scanning microscopy, were applied to study the interactions of this in vitro biofilm community. Three different models of dual-species infections were established in this study to identify biofilm-favoring conditions. In the first two sequential models, a single species of either *E. faecalis* or *P. gingivalis* was seeded prior to its colonization by the second microorganism. The co-culture model, as postulated to be more likely to occur in combined endodontic-periodontal lesions, involved the common contact of both species that shared and cross-infected two habitat niches, i.e., the root canal system and periodontal pockets. All models considered both anaerobic (anoxic) and aerobic (normoxic) conditions.

It was found by next-generation sequencing methods that the key genera in primary and persistent infected root canals included *Prevotella, Fusobacterium, Porphyromonas, Parvimonas,* and *Streptococcus* and the most abundant phyla were *Firmicutes, Bacteroidetes, Proteobacteria, Actinobacteria,* and *Fusobacteria* [28,29]. Using the checkerboard DNA-DNA hybridization method, root canal and periodontal niches were found to share similar microbial profiles [18], and Gram-positive *E. faecalis* and Gram-negative *P. gingivalis* were detected in both root canals with secondary endodontic involvements and periodontal pockets in primary periodontal lesions [18]. A recent scientific report highlighted that the obligate anaerobic *P. gingivalis* could survive in the presence of oxygen when grown along with *C. albicans*, a facultative yeast-like anaerobe [22]. It was shown that *C. albicans* grew in their filamentous form, i.e., hyphae, and this could create a microenvironment with reduced oxygen tension and support the growth of *P. gingivalis* [22]. Obligate anaerobes usually do not possess any anti-oxidation mechanisms like superoxide dismutase to detoxify the oxygen challenge [19]. *C. albicans* is able to create a reduced redox potential environment to support the growth of some anaerobic bacteria, and a gradient of oxygen concentration down to ca. 4% has been observed in C. albicans-formed biofilms [30,31]. On the other hand, *P. gingivalis* cells were shown to survive and tolerate low oxygen levels (6–10%), depending on the haemin availability [32]. Haemin dimers binding to the surface of *P. gingivalis* would serve as a catalase-like buffer system to overcome the oxygen stress and reach a steady-state growth [19,20,33]. Whether *E. faecalis* possesses the same ability to form a protective “biofilm” to shield anaerobic *P. gingivalis* in the aerobic environment has never been investigated.

PCR is usually used to qualitatively detect the presence or absence of a particular bacterial DNA, while it does not provide quantification of the bacterial species of interest and is unable to differentiate live from dead cells [34]. To counteract these issues, quantitative PCR was used in this study to quantify the relative amount of individual living bacterial cells and propidium monoazide (PMA) stain (PMAxx™, Biotium, CA, USA) was used to detect the viable DNA. PMAxx™ dye is a photoreactive dye that preferentially binds to the dsDNA [35]. Visible blue light by PMA-Lite LED Photolysis Device will induce a photoreaction of the chemical, leading to a covalent bond with PMA and the dsDNA. PMAxx™ dye is designed to be a cell membrane-impermeant [36]; thus, only dead cells are susceptible to DNA modification due to compromised cell membranes. Thus, this unique feature of PMAxx™ is highly useful in selectively detecting live bacteria by subsequent qPCR.

CLSM, a non-destructive method often used to investigate the biofilm ecosystem and the hydrated spatial arrangement at the cellular scale, was employed in this study to assess biofilm thickness and relative protein contents [15]. This technique allows for three-dimensional reconstruction of the biomass to study the biofilm architecture [15]. SyPRO biofilm matrix stain was used in this study to allow the visualization of various protein molecules produced by the bacterial species, including glycoprotein, lipoprotein, phosphoprotein, calcium-binding protein, and fibrillar protein [37].

It was reported that *C. albicans* formed filamentous hyphae that facilitate the growth of *P. gingivalis* under the oxic challenge [22]. Similarly, another Gram-negative obligate anaerobe, *F. nucleatum,* could support the growth of *P. gingivalis* in the oxygenated environment by its ability to metabolize oxygen [19]. *F. nucleatum* expresses nicotinamide adenine dinucleotide phosphate (NADPH) oxidase to break down the oxygen molecule [38]. In this study, *P. gingivalis* was found to maintain its viability in the oxic condition in the presence of *E. faecalis*. Hence, the authors proposed that *E faecalis* could probably provide some barrier to protect the obligate anaerobe *P. gingivalis* from the oxygen challenges so that *P. gingivalis* was able to grow and proliferate in the presence of high oxygen tension. The possible mechanism that *E. faecalis* could act as an “oxygen reducing bacteria” is that it is a facultative anaerobe that can grow and utilizes the oxygen present in the biofilm micro-environment [39,40]. Additionally, *E. faecalis* can form a biofilm, and the extracellular polysaccharide (EPS) formed in the biofilm can protect the resident *P. gingivalis.* The EPS also offers protection against various environmental stresses such as pH shifts, osmotic shock, UV radiation, and desiccation [15]. However, a 15-species polymicrobial biofilm model showed that when *E. faecalis* was added to the basic biofilm that consisted of *P. gingivalis* and others, the total cell number was reduced [41]. This could be due to the antagonistic effect of the remaining bacteria present in the biofilm models.

Biofilm thickness was significantly greater when both species were co-cultured simultaneously, regardless of the oxygen challenge. This was further supported by the findings of increased protein contents in the co-cultured biofilms. Biofilm formation is associated with quorum sensing, which regulates bacterial gene expression in response to large cell population densities. A molecule called “autoinducers” was involved in biofilm formation [42]. *E. faecalis* biofilm formation is regulated by the fecal streptococci regulator (*fsr*) locus [43]. Enterococcal cells do not only communicate through Fsr quorum sensing but are also capable of communicating by peptide pheromones, including Cpd, Cob, and Ccf [44,45]. On the other hand, *P. gingivalis’s* capacity to form biofilms is related to the caseionlytic proteases (Clp), which upregulate the expression levels of fimA, mfa1, and luxS [46]. *E. faecalis*’ biofilm development was influenced by growth medium and nutrient availability [16]. In this study, TSB broth in Pg broth contained 3 g of glucose which helped to promote *E. faecalis*’ biofilm formation. It was reported that *E. faecalis* showed highly aggregated biofilms with higher bacteria counts and EPS bio-volume when the medium contained sucrose compared to the medium containing glucose or without sugar substrate [16].

*E. faecalis* possesses many virulence factors, such as enterococcal surface protein (*esp*), gelatinase (*gelE*), aggregation substance (*asa*1), cytolysin B (*cylB*), and endocarditis-specific antigen A (*efaA*) gene, ArgR family transcription factor (*ahrC*), endocarditis and biofilm-associated pili (*ebpA*), enterococcal polysaccharide antigen (*epal*), epal and OG1RF_11715 (*epaOX*), and (p)ppGpp-synthetase/hydrolase (*relA*) genes [47,48,49,50,51,52,53,54,55]. Meanwhile, *P. gingivalis* also possesses various virulence factors, such as lipopolysaccharide (LPS), gingipains, fimbriae/pili, collagenase, (erythrocyte) lectins, capsule, protease, and superoxide dismutase [56,57]. These virulence factors are predominantly proteins of different types, which might have attributed to the higher protein contents observed and thicker biofilm formation in this study when both microorganisms were grown together. A previous dual-species biofilm proteomic study consisting of *F. nucleatum* and *P. gingivalis* showed that more proteins were found when they were co-cultured together as compared with *P. gingivalis* as mono-culture but to a lesser extent than *F. nucleatum* mono-culture biofilm [58]. The reasons were not fully elucidated. It might be due to the antagonist effect of *P. gingivalis* upon *F. nucleatum*. To date, no studies have characterized the proteome of dual-species biofilm composed of *E. faecalis* and *P. gingivalis*. However, some facultative anaerobes such as *C. albicans* and *Streptococcus gordonii* (*S. gordonii*) have been shown to co-adhere to *P. gingivalis* in co-culture [22,59]. The co-adhesion between *P. gingivalis* and *S. gordonii* might have been mediated by the SspB protein of *S. gordonii*, whereas Mp65 adhesin and enolase might have played a role in the interaction between *C. albicans* and *P. gingivalis* (Chung et al., 2000, Bartnicka et al., 2019). It was postulated that the cross-talk between these proteins might have rendered the *P. gingivalis* in the dual-species biofilm the protective mechanism to thrive in an oxic environment. *E. faecalis*, a facultative anaerobe like *S. gordonii* and *C. albicans*, might have also cross-talked with *P. gingivalis* when they were co-cultured, resulting in an increased protein production. Chung and co-workers reported that 100-kDa protein of *P. gingivalis* interacted with *E. faecalis* to induce the cellular surface’s expression of SspB [59]. Furthermore, *E. faecalis* had been shown to take up heme and synthesize heme proteins, called hemoproteins [60]. The hemin content in the culture medium used in this study could have potentiated the hemoprotein synthesis, contributing to the increased protein contents when *E. faecalis* cells were present in the co-cultured biofilms.

In future studies, scanning electron microscopy or fluorescence in situ hybridization (FISH) combined with CLSM could be used to determine the spatial distribution of the bacteria in the biofilm [61], which may reveal the actual architecture of the dual-species biofilm. Furthermore, biofilm protein quantification assay, such as Lowry protein assay or fluorescein isothiocyanate (FITC), can be considered to quantify the protein component in the biofilms [62,63].

## 5. Conclusions

Within the limitations of this in vitro study, it may be concluded that *E. faecalis* appeared to shield *P. gingivalis* from oxygen challenges and support its continued growth in an oxic environment. When both *E. faecalis* and *P. gingivalis* were co-cultured simultaneously, thicker biofilms were formed under both aerobic and anaerobic conditions. A greater amount of protein was also found in the dual-species biofilm model when they were co-cultured aerobically, compared to the mono-culture and sequential culture models.

## Figures and Tables

**Figure 1 microorganisms-10-01729-f001:**
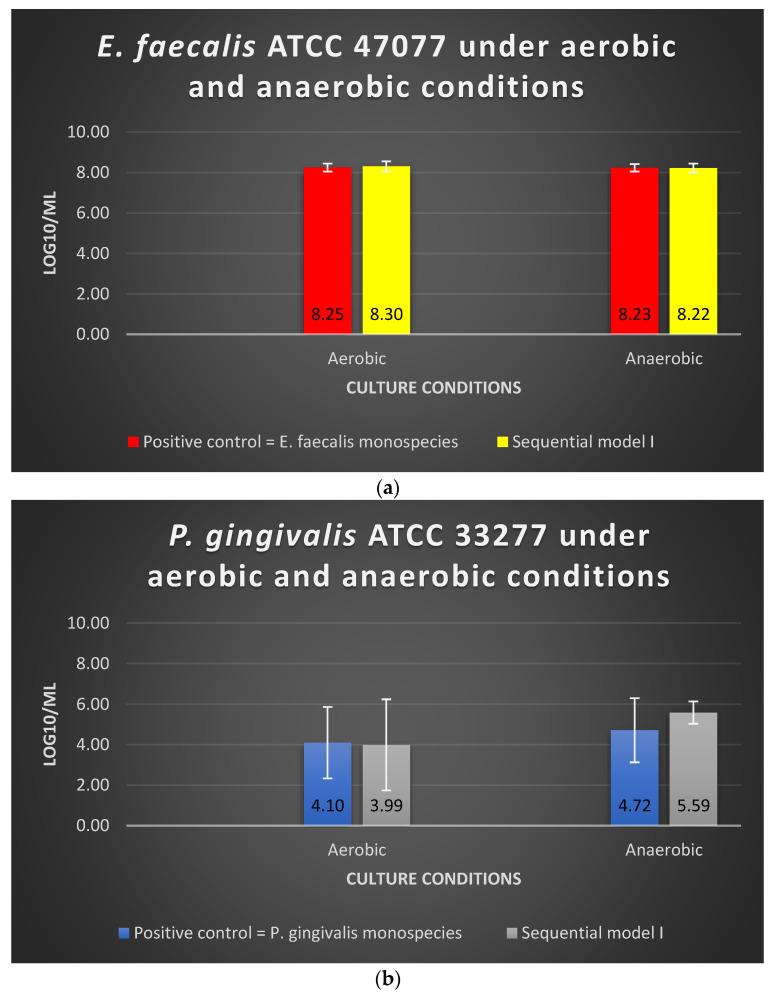
The viability of (**a**) *E. faecalis* and (**b**) *P. gingivalis*, cultured under aerobic and anaerobic conditions based on the Sequential model I and mono-culture models. Data represent the mean and standard deviation.

**Figure 2 microorganisms-10-01729-f002:**
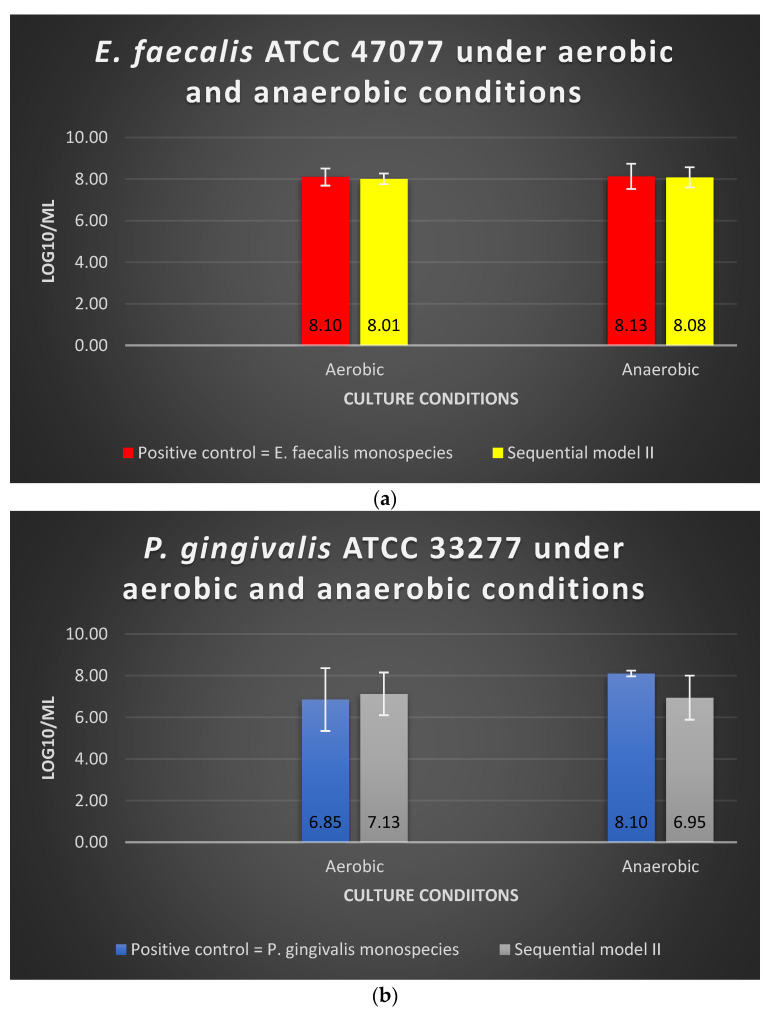
The viability of (**a**) *E. faecalis* and (**b**) *P. gingivalis*, cultured independently under aerobic and anaerobic conditions based on Sequential model II and mono-culture models. Data represent the mean and standard deviation.

**Figure 3 microorganisms-10-01729-f003:**
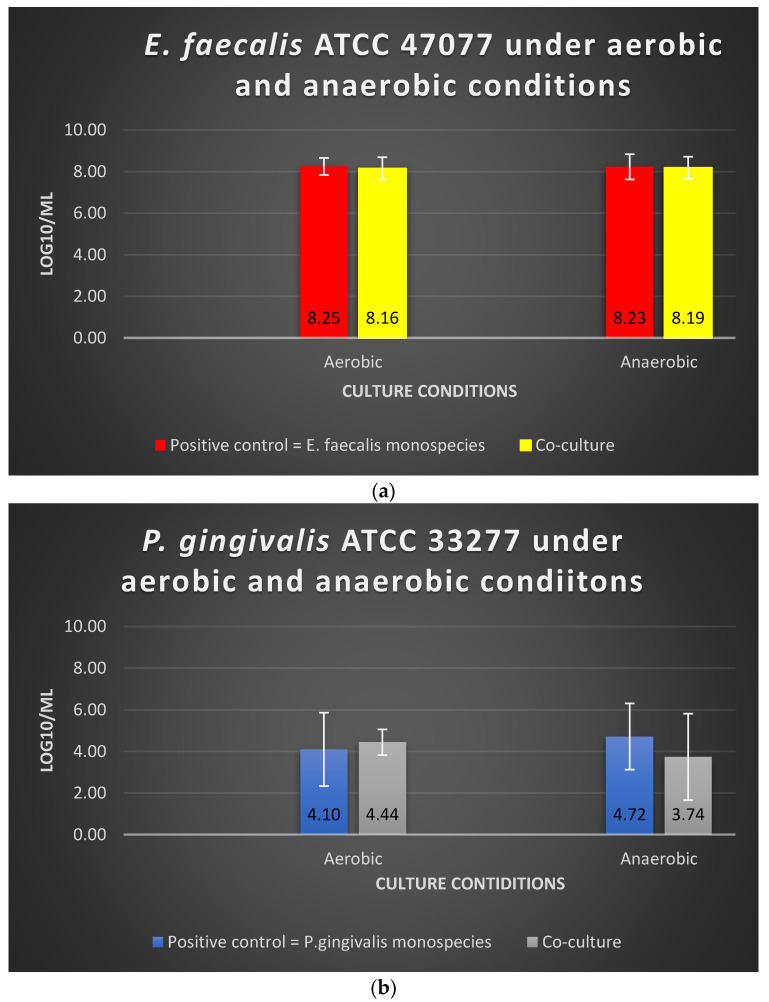
The viability of (**a**) *E. faecalis* and (**b**) *P. gingivalis*, cultured independently under aerobic and anaerobic conditions based on the co-culture and mono-culture models. Data represent the mean and standard deviation.

**Figure 4 microorganisms-10-01729-f004:**
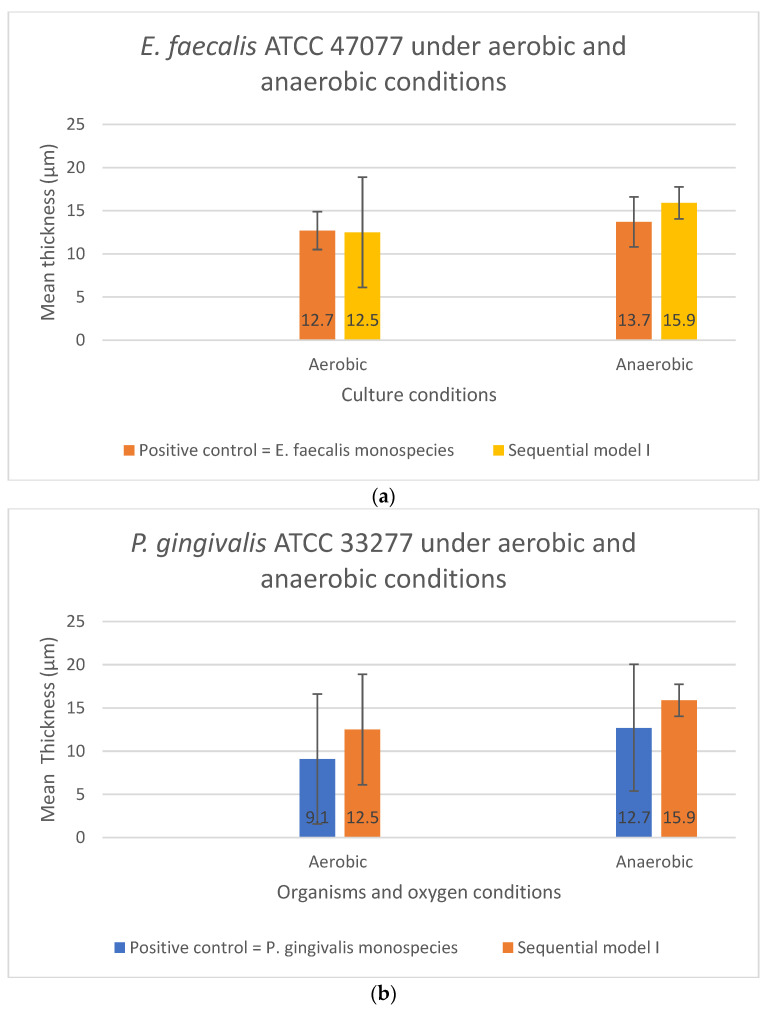
The biofilm thickness of (**a**) *E. faecalis* and (**b**) *P. gingivalis* cultured independently under aerobic and anaerobic conditions based on Sequential model I and mono-culture models. Data represent the mean and standard deviation.

**Figure 5 microorganisms-10-01729-f005:**
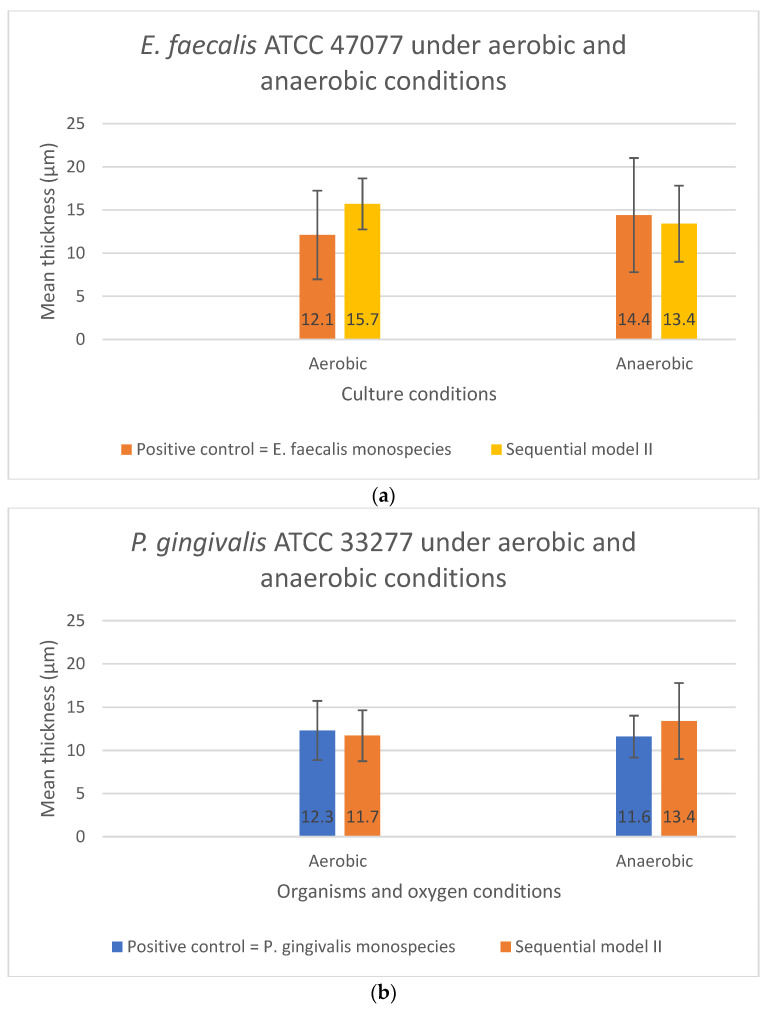
The biofilm thickness of (**a**) *E. faecalis* and (**b**) *P. gingivalis* cultured independently under aerobic and anaerobic conditions based on the Sequential model II and mono-culture models. Data represent the mean and standard deviation.

**Figure 6 microorganisms-10-01729-f006:**
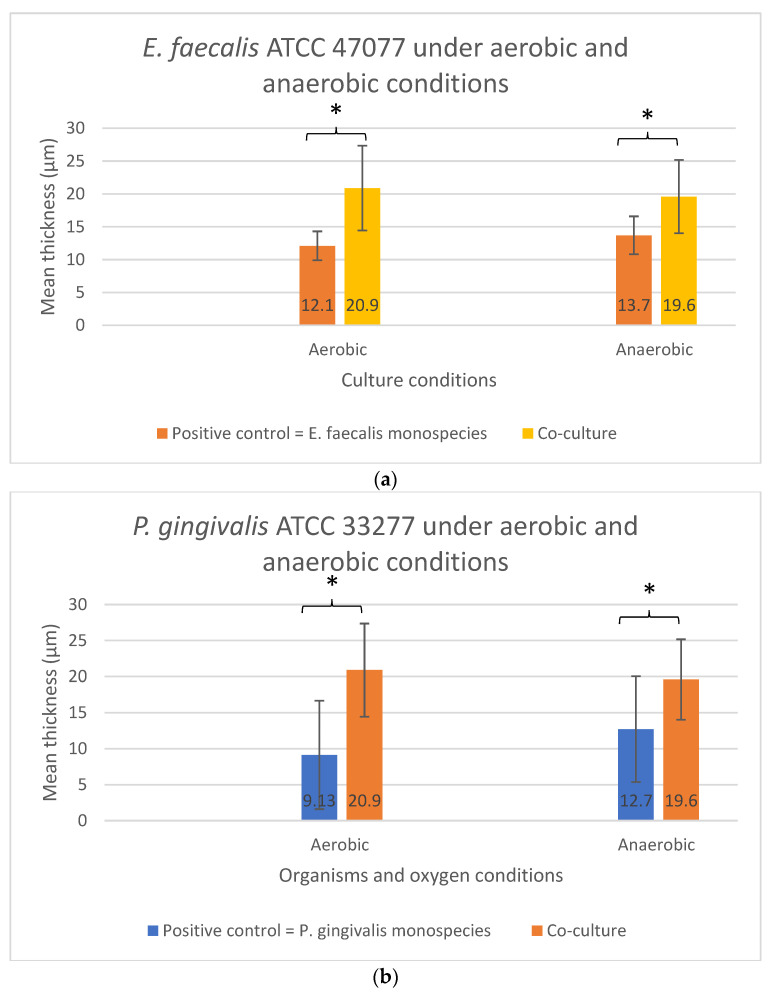
The biofilm thickness of (**a**) *E. faecalis* and (**b**) *P. gingivalis* cultured independently under aerobic and anaerobic conditions based on the co-culture and mono-culture models. Asterisks (*) denote a statistically significant difference (*p* < 0.05). Data represent the mean and standard deviation.

**Figure 7 microorganisms-10-01729-f007:**
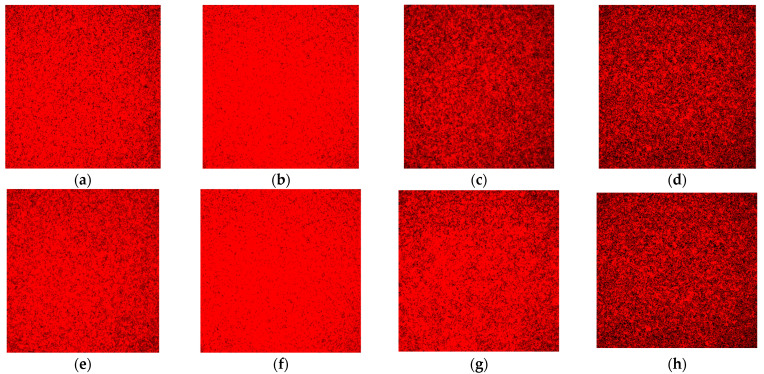
Representative CLSM images of protein content in the biofilm stained with the SyPRO biofilm matrix stain: (**a**) *E. faecalis* as a monospecies culture under aerobic conditions; (**b**) *E. faecalis* as co-culture under aerobic conditions; (**c**) *E. faecalis* as a monospecies culture under anaerobic conditions; (**d**) *E. faecalis* as co-culture under anaerobic conditions; (**e**) *P. gingivalis* as a monospecies culture under aerobic conditions; (**f**) *P. gingivalis* as co-culture under aerobic conditions; (**g**) *P. gingivalis* as a monospecies culture under anaerobic conditions; and (**h**) *P. gingivalis* as co-culture under anaerobic conditions. Note: Protein contents are stained red. Images with a greater degree of redness denote a greater amount of protein contents, as seen in (**b**,**f**).

**Table 1 microorganisms-10-01729-t001:** (a) Species-specific primer sequences used in this study; and (b) Species-specific DNA probe sequences used in this study.

Organisms and Strains	(a) Primer Sequences (5′ → 3′)	(b) DNA Probe Sequences (5′ → 3′)
*Enterococcus faecalis* (ATCC 47077)	Forward: GTTTATGCCGCATGGCATAAGAG Reverse: CAGGTCGGCTATGCA	6FAM-CGGCTCACCAAGGCCA-TAMRA
*Porphyromonas gingivalis* (ATCC 33277)	Forward: ACCTTACCCGGGATTGAAATG Reverse: CAACCATGCAGCACCTACATAGAA	6FAM-ATGACTGATGGTGAAAACCGTCTTCCCTTC-TAMRA

## Data Availability

Data may be obtained from the first and corresponding authors—H.C.T., C.Z. and A.H.C.L.

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
