# Peer review of "Enterococcus faecalis Shields Porphyromonas gingivalis in Dual-Species Biofilm in Oxic Condition"

_microorganisms, 2022, doi:10.3390/microorganisms10091729_

Round 1

Reviewer 1 Report

In this work, the authors have developed a reproducible biofilm model consisting of Enterococcus faecalis and Porphyromonas gingivalis and to evaluate the interaction between the two bacterial species. It is an interesting work but it must be revised to be accepted: 

-In the figure 1a, in the legend, the color of the positive control must be changed to red.

-In all the figures the standard deviation must be appear.

Author Response

Dear Reviewer,

Thank you for reviewing our manuscript and providing your esteemed comments on it.

Comments by you:

  1. In the figure 1a, in the legend, the color of the positive control must be changed to red.
  2. In all the figures the standard deviation must be appear.

Responses to the comments:

We have revised all figures’ legends colors and standard deviations are now included into all the figures presented in this manuscript.

We highly appreciated all the comments given by the reviewers. Please do not hesitate to contact us if any further revision is needed for this manuscript. Thank you.

Best regards

Angeline Lee (corresponding author on behalf of all authors)

Reviewer 2 Report

In this study, the authors investigated the interaction of Enterococcus faecalis and Porphyromonas gingivalis in aerobic and anaerobic condition.

They concluded that E. faecalis appeared to support P. gingivalis against oxygen. When they were co-cultured simultaneously, they formed ticker biofilm in aerobic and anaerobic conditions.

Study is interesting. The points below should be improved.

-       High resolution of Figure 7 should be used. It is not clear what the authors want the reader to see.

-       This is an interaction study; it would be better to see the bacteria under microscope. When the bacteria co-cultured, is there any morphologic change on P. gingivalis or E. faecalis? Electron microscopic studies may be helpful.

Author Response

Dear Reviewer

Thank you for reviewing our manuscript and providing your esteemed comments on it.

Comments by you:

  1. High resolution of Figure 7 should be used. It is not clear what the authors want the reader to see.
  2. This is an interaction study; it would be better to see the bacteria under microscope. When the bacteria co-cultured, is there any morphologic change on P. gingivalis or E. faecalis? Electron microscopic studies may be helpful.

Responses to the comments:

  1. We have revised Figure 7 by replacing all the images with those of better quality. Explanation was also added to the Figure 7’s legend to enhance clarity – “Protein contents were stained red. Images with greater degree of redness denotes greater amount of protein contents, as seen in (b) and (f)”.
  2. Unfortunately, we did not perform any electron microscopic studies when the experiments were conducted; but would certainly include in the future studies as mentioned in the last paragraph of the discussion - “In future studies, scanning electron microscopy or fluorescence in situ hybridization (FISH) combined with CLSM could be used to determine the spatial distribution of the bacteria in the biofilm [61], which may reveal the actual architecture of the dual-species biofilm.”

We highly appreciated all the comments given by the reviewers. Please do not hesitate to contact us if any further revision is needed for this manuscript. Thank you.

Best regards

Angeline Lee (corresponding author on behalf of all authors)

Round 2

Reviewer 2 Report

- The P values of comparisons must be added in each figure's legend. It is difficult to see which comparison is significant, which one is not. Just a suggestion, the author may add asterisk on columns. Then they can mention what the mean of it in each legend.

- I understand that the authors used four replicates for each column in the figures. There are many columns in figures there is no SD. Does the author mean each values of replicates in those columns are exactly same? How is it possible?

- How many times did the author do the each experiment to confirm their results? It must be added to material-method section.

Author Response

Responses to Reviewers’ Comments

Thank you once again for reviewing our manuscript and providing your esteemed comments on it.

Comments by reviewer 2:

  1. The P values of comparisons must be added in each figure's legend. It is difficult to see which comparison is significant, which one is not. Just a suggestion, the author may add asterisk on columns. Then they can mention what the mean of it in each legend.

Response: Thank you for your comment and suggestion. An asterisk (*) has been added to the column that showed a significant difference (p < 0.05), as per your suggestion. Furthermore, “… Data represent the mean and standard deviation” was added to figures 1-6’s descriptions.  

  1. I understand that the authors used four replicates for each column in the figures. There are many columns in figures there is no SD. Does the author mean each values of replicates in those columns are exactly same? How is it possible?

Response: Thank you for your comment. Standard deviations have been added as error bars in figures 1-6.

  1. How many times did the author do each experiment to confirm their results? It must be added to the material-method section

Response: The statement “… All experiments were repeated on three independent occasions” have been added in the materials and method section to improve clarity.

Please do not hesitate to contact us if you have any further questions or comments.

Once again, thank you.

Yours sincerely,

Dr LEE Angeline H.C.
Corresponding author on behalf of all authors